# Direct Impact of the Air on Mutant Cells for Mutagenicity Assessments in Urban Environments

**DOI:** 10.3390/microorganisms12010003

**Published:** 2023-12-19

**Authors:** Chiara Caredda, Elena Franchitti, Giorgio Gilli, Cristina Pignata, Deborah Traversi

**Affiliations:** Department of Public Health and Pediatrics, University of Torino, piazza Polonia 94, 10126 Torino, Italy; chiara.caredda635@edu.unito.it (C.C.); elena.franchitti@unito.it (E.F.); cristina.pignata@unito.it (C.P.)

**Keywords:** mutagenicity, Salmonella assay, air pollution, urban environment

## Abstract

Background: Urban air pollution is recognized as a critical problem for public health and is classified as a carcinogen for humans. A great number of studies have focused on the monitoring of urban air mutagenicity. One of the best-known and applied methods for assessing mutagenicity is the Ames test, a bacterial reverse mutation test. The classic protocol for assessing air mutagenicity involves the concentration of particulate matter (PM) on filters and subsequent extraction using organic solvents. This work aimed to develop a method for the evaluation of air mutagenicity directly impacted by air on microbial plates already containing an Ames’ microbial sensor. Methods: A specific six-month sampling campaign was carried out in Turin in a period with high air pollution. Samples were tested for mutagenicity on *Salmonella typhimurium* strains TA98, TA100, and YG1024 with the traditional method and with the new direct method. Results: The new protocol is able to evaluate the mutagenicity of the sampled air and obtain repeatable results. The final sensitivity is similar to the traditional method (≈10 net revertants/m^3^); however, the mutagenic response is due to the complete air pollution mixture, including volatile and semivolatile pollutants avoiding the concentration of filters and the following laborious extraction procedures. Conclusions. Despite some critical issues in contamination control, the method is easier, faster, and less expensive than traditional methods.

## 1. Introduction

The continuous increase in population density, especially in urban areas, is associated with an increase in air pollution [1,2,3]. Air is a complex matrix that includes many gaseous elements and particles. Particulate matter (PM) refers to air pollution particles composed of micro-fragments. It is a complex mixture of both suspended solid particles and liquid droplets which vary in size (PM_10_, PM_2.5_, PM_0.1_) and composition [4].

The level of air pollution today is lower in quantity than in the past, and it is also different in terms of physical-chemical quality [5]; however, because of chronic exposure to already high levels of air pollutants, some diseases of the respiratory tract, such as asthma, bronchitis, emphysema, and cancer, are clearly promoted. There is a lot of scientific evidence showing that, in urban areas, there is a higher incidence of these diseases, especially cancer of the respiratory tract [6]. Ten years ago, the IARC classified air pollution as a group 1 agent [7]. Air particulate matter pollution was the second-highest specific risk factor for deaths from tracheal, bronchial, and lung cancer, accounting for 15% of tracheal, bronchial, and lung cancer mortalities worldwide. For each 10 μg/m^3^ increase, of the air pollutant criteria, lung cancer mortality increased by up to 30% [6]. People with heart or lung disease, children, and the elderly are among the categories considered to be at higher risk. This has an economic impact on public health. The WHO periodically publishes health-based air quality guidelines to help governments and civil society reduce human exposure to air pollution and its adverse effects [3]. This has current economic and juridical consequences for the regulatory institution and has initiated various citizen class actions in cities in France, Germany, and Italy against the government for health problems linked to air pollution.

A reduction in air pollution, and the mitigation of its adverse effect on human health, is mandatory both in Europe and in extra-European countries [8]. In this context, the monitoring of air pollution and an accurate assessment of its potential effects are determinants for public health purposes. Current methods include the monitoring of some air pollutants, codified, for example, by the European Directive 50/2008, such as PM_10_, PM_2.5_, NO_x_, SO_2_, and O_3_; however, these only include the sporadic effect of air pollution on biosensors.

The most commonly used methods for the study of air pollution interactions with DNA are mutagenicity and genotoxicity tests [9,10]. Many substances present in the air induce damage at the genetic (from point mutations to chromosomal mutations) and epigenetic levels, which can be the basis for the development of human cancer [11]. The evaluation can include an approach based on laboratory tests on biosensors [12] or human biomonitoring [13]. The laboratory evaluation generally started from PM_10_ or PM_2.5_ collection. The particles contain the more mutagenic and genotoxic substances present in the air; for this reason, mutagenicity and genotoxicity tests are frequently performed starting from organic extracts of such fine particles. There is a large body of literature that shows the mutagenicity of air pollution, and a wide range of studies have been carried out on every continent [10,11,14,15]. In Italy, in recent years, the Regional Environmental Protection Agency (for example, Emilia Romagna and Piemonte) applied the Ames test to determine the mutagenicity of PM_10_ and/or PM_2.5_, translating the research evidence into a monitoring system [12,16,17,18].

However, various methods limit mutagenicity test inclusion in routine monitoring, among which, (1) during air pollution collection and following organic extraction, volatile and semi-volatile compounds are lost; (2) the concentration on the filter and subsequent organic extraction can modify the characteristics of original air pollutants through a secondary transformation; (3) the assay is time-consuming and/or quite expensive; (4) the results cannot be obtained within 48 h; (5) the results are generally strictly correlated to PM levels that are missing to provide an additional input for health risk assessments.

This work aims to develop a method for the evaluation of air mutagenicity through the direct impact of air on microbial plates already containing cells of genetically modified *Salmonella typhimurium*.

## 2. Materials and Methods

### 2.1. Sampling Settings

Two different sampling methods (traditional PM_10_ sampling on filters and the direct system on microbial plates) were carried out from October 2021 to January 2022 during the winter season in an urban background area of Turin. Turin is one of the most polluted cities in southern Europe [18,19,20,21] and it is located in the northwest of the Po Valley, where air pollution is a critical issue [5].

The choice of the sampling period is due to the higher air pollution observed during this season in the past [19]. Winter seasons, in such areas, are favorable to the accumulation of airborne pollutants (low temperatures, less dispersion, thermic inversion influence, and household heating), accounting for more than 100 critical days during both 2021 and 2022 [22,23]. Therefore, it seems that, in winter, the agglomeration of ultrafine particles into larger particles (especially in the 70–100 and 100–200 nm ranges) and the condensation of semivolatile gases on pre-existing particles are promoted as a consequence of low temperature, the high concentration of airborne pollutants, and reduced pollution dispersion [24]. In the winter period, civil and industrial heating is added to the usual sources of traffic, producing pollutants, including nitrogen compounds and sulfur oxides. These last events during winter in the Po Valley give rise to nitrate and ammonium sulfate in the particulate phase. In 2020, the month in which particles larger than 100 nm represented the majority compared to the total was January [23].

The sampling site was a meteorological-chemical station of ARPA Piedmont called Lingotto [18]. It is located within a green area near relevant road crossings. Part of the fenced external space of the station was used to perform the air sampling, and 12 sampling days were performed with the two sampling systems in parallel (Table 1). During each sampling day, PM_10_ samplings were performed simultaneously using the Airflow PM_10_ instrument, which is necessary for the execution of the classic Ames test after organic extraction of the filter, and direct plate sampling via DUO SAS Super 360 for the direct assessment method.

### 2.2. Direct Method Development

The *Salmonella typhimurium* strains included in this test were all histidine-dependent and had rfa, uvrB/A, and pkM101 modifications [25]. These were *Salmonella typhimurium* TA98 and TA100 (commonly used in Ames tests as a model of frame-shift mutation and base substitution, respectively) and YG1021 and YG1024, which have undergone other genetic modifications to be even more sensitive to air mutagens (nitroreductase or O-acetyltransferase-overproducing, respectively) [26].

Before the sampling campaign, preliminary tests were carried out (1) to evaluate the effectiveness of the protocol in the smallest contact plate, (2) the effective growth capacity of the selected *Salmonella typhimurium* strains (TA98, TA100, YG1024, YG1021) in the culture medium after the air impact on the plate, (3) the resistance of the strains to the contemporary addition to the medium of substances for bioaerosol growth inhibition, (4) the positive control for the action of known mutagens, and (5) the positive control for the action of a diesel exhaust mixture sampled directly on the plates.

The same base media used in the classical method, including 0.05 mM Histidine–Biotin Solution, were prepared for the contact plates (RodacTM Contact Plates, 55 mm, VWR, USA). A detailed description of the protocol phases is given in the Appendix A.

Generally, the mutagenic potential of the samples is also evaluated in the presence of metabolic activation provided by a liver homogenate (S9 mix) [10]. In this work, only the direct mutagens were evaluated because, in the new method, the S9 mixed addition was not practicable at the same time as the air impact on the Salmonella cells.

For the direct method, air sampling on the plates was carried out using DUO SAS Super 360 (VWR International s.r.l., Milan, Italy) on a support approximately 1.7 m from the ground. Three increasing volumes were selected for each strain of *Salmonella typhimurium* in triplicate. The sampling duration is short and varies depending on the selected air volume from a few minutes (100 L) to 15–20 min (2000 L). At the end of the sampling, the plates were placed back into the laboratory and positioned in the incubator for 48 h at a temperature of 42 °C.

### 2.3. Traditional PM_10_ Sampling and Extraction

PM 10 sampling was performed for the classic Ames test using a high-volume sampler, AIRFLOW PM_10_ (AMS Analitica Air Sampling System, Pesaro, Italy), as set out in the European standards [27] and in compliance with the international standard [28]. Briefly, this method involves selections made through an impactor system and the collection of PM_10_ using filters. The instrument is set to carry out sampling for a duration of 24 h, with a flow rate of 1.27 m^3^/min. The filters, in glass fiber 600/GA55 (PALL, Cortland, NY, USA), with a diameter of 8 × 10 inches, were subjected to a conditioning treatment before and after the sampling and the gravimetric analysis to eliminate humidity using a dryer containing silica gel for 48 h in a dark environment. PM_10_ organic extract was obtained using acetone and Soxhlet apparatus for at least 80 cycles [21,29]. The extract was then dried using a Rotavapor and resuspended in DMSO to obtain a concentration of 0.4 m^3^ equivalent to sampled air/µL.

The Ames test was performed following the guidelines [25] with the exception of the condition detailed in Table 1, which was needed for the comparison of the two methods. In this case, there is no direct exposure of the plates to the air; however, the application was carried out on the plates of the PM organic extracts located at −20 °C until the classic Ames test was performed. The plates were then prepared (100 mm diameter classic microbiology plates). Different concentrations of the extract (2 µL, 10 µL, 20 µL, and 50 µL) for each filter belonging to each sampling day were added to the plate. Subsequently, 2.5 mL aliquots of soft agar and 100 µL of one of the three *Salmonella typhimurium* strains (TA98, TA100, YG1024) were added to the various test tubes. This was prepared for each strain in triplicate for each dose of the extract. Positive and negative controls were prepared. Then, the plates were incubated at 42 °C for 48 h before the results could be read.

### 2.4. Data Analysis and Statistics

After incubation, colony-forming units (CFUs) on the plate were observed. The slope of the dose–response curve (revertants/m^3^) was calculated using the least square’s linear regression from the linear portion of the dose–response curve [19]. All experiments were performed in triplicate with at least three doses. The results were expressed as total revertants minus spontaneous revertants to obtain net revertants per cubic meter (rev/m^3^) and were calculated using the dose–response curve.

Additional data on PM_10_, PM_2.5_, and black carbon monitoring for the same sampling day and the same air monitoring station were also extracted from a database provided by the Regional System for the real-time monitoring of air quality in Piedmont (https://aria.ambiente.piemonte.it/#/by ARPA Piemonte).

Statistical analyses were performed using the IBM SPSS Statistics, version 28.0 (IBM Corp., Armonk, NY, USA). In particular, we applied a Mann–Whitney nonparametric comparison test for independent samples, a nonparametric Kruskal–Wallis one-way ANOVA for comparing (number, *n* > 2) independent samples, and the Spearman rank order correlation coefficient to assess relationships between variables. The results were considered statistically significant when the *p* value was <0.05 and highly significant when the *p* value was <0.01.

## 3. Results and Discussion

The air is obviously contaminated by environmental microorganisms, and the growth inhibition of the non-*Salmonella typhimurium* strain is a crucial issue. First, some considerations must be considered: (1) the base agar involved in plate preparation is a poor medium for the growth of the majority of microbes; (2) the test included antibiotics (ampicillin, Sigma Aldrich, Milan, Italy), and the genetic modifications of *Salmonella typhimurium* include pkM101 (plasmid that confers resistance to ampicillin); (3) the addition of cycloheximide (Sigma Aldrich, Milan, Italy) is included for fungal control. However, during the preliminary tests, external contamination has been observed and evaluated, and various approaches have been attempted for their elimination.

The effective modifications of the protocol were as follows: (1) increasing the temperature of incubation (from 37 to 42 °C) resulted in an important contamination-inhibiting effect for the environmental microorganisms, (2) increasing the number of Salmonella cells included in the plate to optimize the mutation ratio after exposure, and (3) increasing the number of replicates for each dose (at least 4 plates) to exclude a plate if it was eventually contaminated.

The problem was not definitively eliminated; however, contamination was observed sporadically, including colonies of *Bacillus cereus*, *Pseudomonas luteola* (API test 99%, Biomerieux Italia Spa, Bagno a Ripoli, Italy), and *Sphingomonas paucimobilis* (API test 88%). Such microorganisms were previously identified in the urban aerobiome [30].

An antibiotic (ampicillin 3.15 µL/ml) and an antimycotic (cycloheximide 0.2 g/L) were added as well as the DMSO (dimethyl sulfoxide) organic solvent used to dissolve the mutagenic compounds. Cycloheximide was not able to cause gene mutations or unscheduled DNA repair [31], and such evidence was confirmed by our tests (Appendix A). A first run of the test was then performed without exposure, which had a positive outcome (no toxic effect on the strain can be observed) for TA98, TA100, and YG1024, while toxic effects were observed for YG1021 (approximately 60% of spontaneous revertants) in the presence of cycloheximide. Such widely used eukaryotic protein synthesis inhibitors sometimes also inhibit bacteria [31]. Therefore, the sampling campaign began involving the *Salmonella typhimurium* TA98, TA100, and YG1024 strains. The addition of other antimicrobials (chloramphenicol and/or tetracycline 1.5 mg/100 mL medium) was previously tested but not included in the sampling protocol for significant inhibition (ANOVA *p* < 0.01) (Appendix A), which was not offset by an improvement in the method (in terms of contamination reduction).

Two modifications of the protocol were made to reduce residual bioaerosol contamination. The incubation temperature increased (42 °C), inhibiting the growth of environmental microorganisms but not of the test organisms. Another modification was made to increase the concentration of *Salmonella typhimurium* included in the test (three-fold). The volumes of air sampled with the instrument are 400, 700, and 1000 L, in quadruplicate for each volume. After the modifications, an improvement in the results was clearly obtained, and it was possible to carry out repeatable samplings; moreover, the presence of contamination was strictly reduced (−70% of plates with at least one contamination present).

Salmonella growth was also observed on the RodacTM contact plates after the air impact on the plate (spontaneous revertants/plate). However, it is very low; therefore, the cells seeded into the plate were assayed in triplicate to optimize the observed number of revertants in the incubation conditions selected. Finally, the spontaneous revertants observed were TA98 9 ± 9, TA100 46 ± 28, and YG1024 11 ± 13.

The number of spontaneous colonies of *Salmonella typhimurium* strains observed with the direct test was numerically reduced compared to the data in the literature. For the TA98 strain, spontaneous colonies are generally estimated to be between 20 and 50; for TA100, between 75 and 100; and, for YG104, approximately 30 [26,32]. This could be because, in this method, the strains are subjected to higher stress. In fact, unlike the classic test, incubation (at a higher temperature of +5 °C with respect to the classical protocol) takes place after round-trip transport to the sampling station.

To obtain a real positive control, sampling was carried out with the exposure of the plates using the same DUO SAS Super 360 sampler on the emissions of a heavy vehicle’s diesel engine (EURO 3), a notable mutagenic mixture. With the exposure of 700 L of air sampled in the proximity of the emission point, the following net revertants were observed: 23 ± 5, 58 ± 18, and 132 ± 19 for TA98, YG1024, and TA100, respectively. Such results showed the capability of such a method to detect a mutagen effect.

Twelve filters were used to collect PM_10_, and the average concentration of this fraction of the particulate matter during the sampling period was 42.08 µg/m^3^. The minimum recorded was 8.45 µg/m^3^ (first day of sampling), and the maximum was 89.24 µg/m^3^ (on the third day of sampling in December). In accordance with what is highlighted in the literature, particulate matter is more present in the winter period, with a peak in the months of December and January; meanwhile, it decreases in the months with a milder climate, showing a greater dispersion in particulate pollution.

The mutagenicity potential observed is detailed in Table 2 as a classic potential. The mutagenicity of the PM_10_ organic extract is not very high with respect to the past [29] but is similar to more recent PM_2.5_ mutagenicity observed in the same territory and in a similar period of the year [18].

The direct assay produced a generally limited number of colonies for each strain, indicating a moderate mutagenic effect of the air (Table 2—direct). However, at the same exposition dose, the assay response is similar (Figure 1), and the slopes of the dose–response curve (expressed as net revertants/m^3^) are not different from the classical method applied to the PM_10_ organic extracts of the same days (Figure 2). The observable difference in the application of the direct method is due to a technical problem. It is not possible to overcome 2000 L of direct air exposition while avoiding plate compromise; however, in the traditional method, higher exposition doses are possible and generally applied.

The mutagenicity observed with the TA100 strain is higher than that of the other strains (*p* < 0.05), showing a higher base substitution effect, probably due to the classical reverse mutation of the TA100 strain [33].

Considering the gravimetric analysis, the data produced using ARPA for PM_2.5_ and PM_10_ correlate significantly with each other (Spearman’s rho = 0.988; *p* < 0.001), and a significant correlation was also highlighted between our PM_10_ and both ARPA’s PM_10_ and ARPA’s PM_2.5_ (Spearman’s rho = 0.804 and =0.767, *p* < 0.01, respectively).

Therefore, the results show that our method of evaluating PM_10_ mutagenicity is always strictly correlated with the PM_10_ concentration in the air, which is expected when starting the test with an organic extraction of the same PM_10_. On the other hand, the mutagenic effect determined by the direct test seems to show a different mutation action not attributable to PM pollution.

As shown in Figure 3, the mutagenicity assessed using the direct method is correlated with the PM_2.5_ and PM_10_ levels (Spearman’s rho = 0.562, *p* = 0.029) while the traditional test is more correlated to NOx levels (Spearman’s rho = 0.556, *p* = 0.031).

The mutagenicity is strictly correlated with the PM_10_ and PM_2.5_ levels for both the TA100 and TA98 results (Spearman’s rho = 0.900 *p* < 0.05) but not for YG1024. Moreover, considering the total net revertants/m^3^ observed, mutagenicity was inversely correlated with black carbon (Spearman rho = −0.319, *p* < 0.01). This was not observed for YG1024 mutations. In the literature, a lower sensitivity of TA100 to air pollution, especially without S9, was shown [34]; meanwhile, a correlation of YG1024 revertants with the levels of nitro-PAHs and hydroxylamines in the finest particles was frequently observed [35]. A comparison was made between the results obtained with the developed direct test and with the classic Ames test (Table 2). There were no significant correlations between the results of the mutagenicity observed with the two methods and no difference in the intensity of the mutagenic effects. Additionally, considering the small amount of data and their high variability, the mutagenicity seems to be attributable to different compounds selected by the different protocols.

The particulate phase is not the only mutagenic element present in the air and is dangerous for human health. In fact, the air contains a high number of carcinogenic and genotoxic compounds, varying in relation to local anthropogenic and natural sources, meta-chemical conditions, and air environmental chemistry.

The air quality is generally better than in the past, considering the criteria parameters for the assessment of air quality. However, updated quality guideline values have been further reduced compared to those in a previous report [3].

The carcinogenicity of urban air pollution produces a responsibility impact on the institution invested in human health protection. Moreover, a legal and economic backlash is possible; in fact, in the last year, various class actions have been promoted by associations of citizens in different European countries (Germany, France, and Italy).

In this context, monitoring through biological effect assays seems to be a priority. On the other hand, both the cost and the time necessary to perform the assays, as the high correlation to the PM_10_/PM_2.5_ mass levels, are linked to the uselessness of mutagenicity assessments [36].

However, the physical-chemical quality of PM_10_ or PM_2.5_ can be very different, with the same mass-producing effects. Many modifications of the Ames test have been proposed, focusing mainly on time optimization for test execution [36], and only a few tests have been performed on the direct effect of artificial atmospheres on Salmonella biosensors [15].

The direct test assesses the mutagenicity of the air as a complex mixture, including chemical and biological components and volatile and semivolatile compounds, which are often lost using other analysis techniques.

An advantage of using the direct test is certainly represented by the shorter execution time needed, with further simplification of the protocol thanks to the exclusion of the extraction and concentration phases of the sample, which, consequently, also makes it lowly priced.

The direct test offers the opportunity to avoid organic extraction and the necessities required to start from particulate matter filters. However, contamination control should be further investigated, especially in the presence of higher bioaerosol presence (rural sites or highly impacted sites such as waste treatment plants) [37,38,39].

Another disadvantage is represented by the small volume of air sampled with this technique. Today, for example, ARPA Piemonte performs the overall extraction of all the filters sampled during a whole month, which is obtained through daily sampling lasting 24 h, and represents the monthly mutagen potential [18], as proposed in most of the literature. With the direct test, we obtain a punctual measurement, namely, a sampling with an average duration of 10 min. This could be a disadvantage because there is greater variability, and it is more likely to be influenced by the conditions and by the sampling site itself; however, it could be advantageous when evaluating specific emission sources. Additionally, from a handler’s point of view, the easy portability of the sampler, with respect to the PM_10_/PM_2.5_ environmental samplers, has to be considered. The low flow rate is more comparable to the human breath rate (≈14 L/min) than the other sampling systems, allowing a mutagenic assessment.

## 4. Conclusions

The mutational process that takes place in *Salmonella typhimurium* can be a good method for environmental monitoring and screening. The proposed direct method seems to be a valuable alternative compared to traditional methods. At the moment, it would probably be considered for use in urban contexts, in which there is usually a lower variability of the bioaerosol, as well as due to the greater presence of pollutants that reduce the biodiversity of the aerobiome. A wider sampling campaign and the improvement of the method testing other air pollution contests covering various seasons could be auspicious. Such a method could be a tool in sustainability evaluations, considering both a source’s impact on the total mutagenicity and the improvements observable after environmental management interventions by institutions.

## Figures and Tables

**Figure 1 microorganisms-12-00003-f001:**
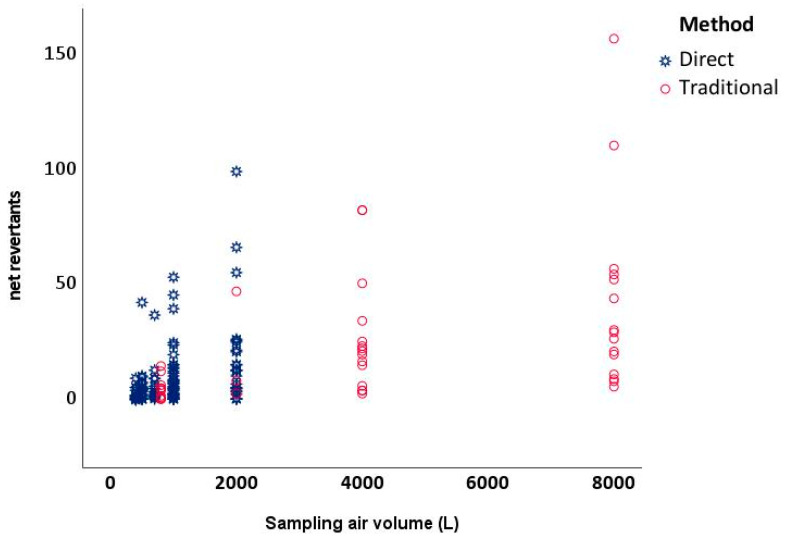
Scatter plot of the overall net revertants observed during the experiments both for the direct methods (three tested volumes) and the traditional methods (first three volumes tested).

**Figure 2 microorganisms-12-00003-f002:**
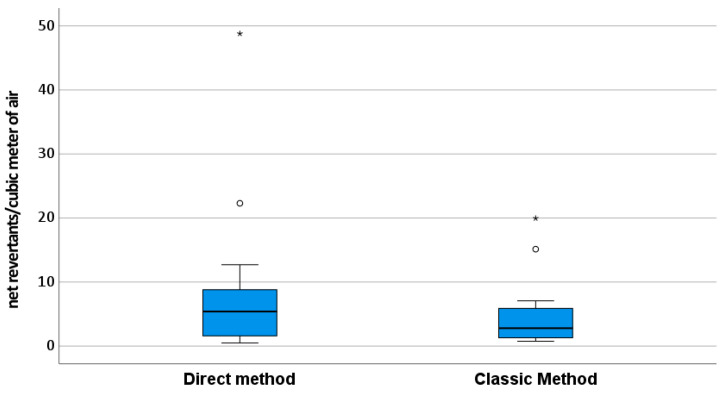
Box plot of the mutagenicity observed (*Salmonella typhimurium* TA98, TA1001, and YG1024 net revertants/m^3^) in the same sampling point and days with the two different methods: the direct method and the classic method. Circles represent the outliers, asterisks the extreme outliers.

**Figure 3 microorganisms-12-00003-f003:**
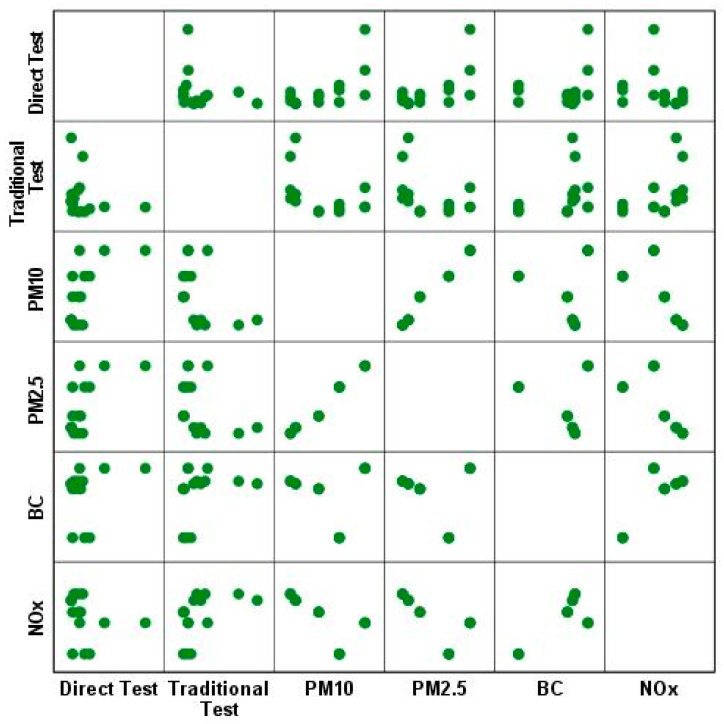
Dispersion matrix of the mutagenicity results obtained using direct and traditional tests with the different collected air pollution parameters.

**Table 1 microorganisms-12-00003-t001:** Main characteristics of the 2 applied tests for the mutagenic assay comparison after protocol development. The common modifications, with respect to the traditional method, are shown in bold for comparison.

	Direct Test	Traditional Test
**Sampling days**	October 6, 12, 20; November 24; December 1, 7, 15; January 11, 12, 18, 19, 25	
**Sampling place**	Turin, Lingotto environmental monitoring station	
**Sampling equipment**	DUO SAS Super 360 sampler (Avantor International)	Airflow PM_10_ instrument (Analitica Strumenti S.p.A.)
**Sampling type**	Direct air impact on plates low flow (180 L/min)	PM_10_ selection on glass fiber filters, high flow (1.27 m^3^/min)
**Plate preparation**	Before the air sampling on RodacTM Contact Plates, VWR, USA (∅ 55 mm)	After the air sampling, microbiological plate (∅ 100 mm)
**Antimicrobials included in the medium**	Ampicillin 2.5 mg/L and Cycloheximide 0.2 g/L	
**Duration of each sampling**	≈2 h in total, from 1 to 3 min for each plate according to the set sampling volume, middle of the day (generally 12:00–15:00)	24 h
**Salmonella introduction on plates**	Just before (≈1 h) the air sampling, 3-fold Salmonella cells included	After the organic extraction of the PM_10_ filters, simultaneously with the inclusion of the organic extract on the plates, a one-fold number of Salmonella cells included
**Salmonella exposition to the sample**	Air pollution as it is	PM_10_ organic extract
**Incubation time and temperature**	48 h, temperature 42 °C	
**Result expression**	Salmonella UFC/plate, then expressed as median and standard deviation and referred to the cubic meter	

**Table 2 microorganisms-12-00003-t002:** Descriptive analysis of the mutagenic results (net revertants/m^3^) for each strain and each method.

Salmonella Strain	Method	Min	Max	Mean	Std. dev.
**TA98**	Direct	1	22.3	8.8	8.7
	Classic	0.8	20	9.2	8.1
**TA100**	Direct	0.5	48.8	14.2	19.7
	Classic	0.8	6.5	2.7	2.4
**YG1024**	Direct	0.8	8.2	3.7	3.3
	Classic	1	20	9.2	8.1

## Data Availability

The dataset generated during the current study is available from the corresponding author upon reasonable request.

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
