# Peer review of "Direct Impact of the Air on Mutant Cells for Mutagenicity Assessments in Urban Environments"

_microorganisms, 2023, doi:10.3390/microorganisms12010003_

Round 1
Reviewer 1 Report
Comments and Suggestions for Authors
In this article, the authors have successfully investigated the direct impact of the air on mutant cells for mutagenicity assessment in an urban environment. This work can be considered for publication after addressing the following issues:
1. In the background section of the abstract, attention should be paid to logical relationships in writing, and it is recommended to readjust.
2. In this article, there are many cases where the numbers in PM10 and PM2.5 have no subscript, please check them carefully. For example, this situation occurs in lines 50, 57, and 64, etc.
3. The article provided a detailed introduction to the hazards of air pollution, but did not describe the sources of air pollution or the classification of air pollution particles. It is recommended to add relevant content, and some references should be added, such as Adv. Mater. 2020, 32, 2002361, Nat. Commun. 2023, 14, 4432, Adv. Funct. Mater. 2019, 29, 1904108, Epidemiol. Health. 2022, 44, e2022071, etc.
4. There are many errors in the citation format of references, please check them carefully. For example, this situation occurs in lines 372, 374, 381, 384, 419 and 428, etc.
5. In line 87, it is mentioned that ultrafine particles will aggregate into large particles in the winter. Please explain the reason for this phenomenon.
Author Response
microorganisms-2760931
We reported the editor/reviewer’s comments and in bold our answers and text changes in the manuscript.
Reviewer 1
In this article, the authors have successfully investigated the direct impact of the air on mutant cells for mutagenicity assessment in an urban environment. This work can be considered for publication after addressing the following issues:
- In the background section of the abstract, attention should be paid to logical relationships in writing, and it is recommended to readjust.
It was revised and re-adjusted
- In this article, there are many cases where the numbers in PM10 and PM2.5 have no subscript, please check them carefully. For example, this situation occurs in lines 50, 57, and 64, etc.
It was changed everywhere in the text
- The article provided a detailed introduction to the hazards of air pollution, but did not describe the sources of air pollution or the classification of air pollution particles. It is recommended to add relevant content, and some references should be added, such as Adv. Mater.2020, 32, 2002361, Nat. Commun.2023, 14, 4432, Adv. Funct. Mater. 2019, 29, 1904108, Epidemiol. Health. 2022, 44, e2022071, etc.
Particle classification was included and the corresponding reference, too.
- There are many errors in the citation format of references, please check them carefully. For example, this situation occurs in lines 372, 374, 381, 384, 419 and 428, etc.
We are using Mendeley software for the bibliography management. I adjusted the references to regulations that were not correct.
- In line 87, it is mentioned that ultrafine particles will aggregate into large particles in the winter. Please explain the reason for this phenomenon.
We added a sentence and a bibliographic reference.
Reviewer 2 Report
Comments and Suggestions for Authors
The authors propose a modified form of the classic Ames test, where airborne pollution is directly interacting with plates containing Salmonella typhimurium strains. This method is presented as a form of interaction of particulate matter with the test organisms, without need of organic extraction and limitations of traditional collecting filters. Disadvantages include a higher risk of microbial contamination and a limitation to rather short sampling times.
The English text of the manuscript is quite good, no major problems have been detected.
The method presented by the authors represents a simplification of the classical Ames test application for the investigation of carcinogenicity/mutagenicity of airborne pollutants. The authors appear to be well aware that the methodical "simplification" is introducing some limitations to the test. A higher degree of uncertainty may be introduced by an overall higher degree of variability and a possibility of secondary microbial contamination, that cannot be fully excluded by temperature adjustemts and suggested antimicrobial treatment. The most important limitation may be given by the rather short maximal exposure time, allowing for only short time sampling and representation of the e temporal variation in air pollution. Fig. 1 also suggests, that larger air volumes (> 2000 l) can only be sampled by the traditional method, and not by the introduced test system.
The method introduced by the authors does however also provide some advantages, like an unmodified and feature-complete representation of air pollution, including solid particles, and may be a valuable alternative to classical sampling and Ames test in specific settings. I therefore support the publication of this manuscript.
ljne 11, Abstract: "Ames test" is maybe more common term than "Salmonella assay" for the abstract.
Supplement: VBC: please explain abbreviation, suggest no capital letters for glucose, nitroflurene sodium azide..
Author Response
microorganisms-2760931
We reported the editor/reviewer’s comments and in bold our answers and text changes in the manuscript.
Reviewer 2
The authors propose a modified form of the classic Ames test, where airborne pollution is directly interacting with plates containing Salmonella typhimurium strains. This method is presented as a form of interaction of particulate matter with the test organisms, without need of organic extraction and limitations of traditional collecting filters. Disadvantages include a higher risk of microbial contamination and a limitation to rather short sampling times.
The English text of the manuscript is quite good, no major problems have been detected.
The method presented by the authors represents a simplification of the classical Ames test application for the investigation of carcinogenicity/mutagenicity of airborne pollutants. The authors appear to be well aware that the methodical "simplification" is introducing some limitations to the test. A higher degree of uncertainty may be introduced by an overall higher degree of variability and a possibility of secondary microbial contamination, that cannot be fully excluded by temperature adjustemts and suggested antimicrobial treatment. The most important limitation may be given by the rather short maximal exposure time, allowing for only short time sampling and representation of the e temporal variation in air pollution. Fig. 1 also suggests, that larger air volumes (> 2000 l) can only be sampled by the traditional method, and not by the introduced test system.
The method introduced by the authors does however also provide some advantages, like an unmodified and feature-complete representation of air pollution, including solid particles, and may be a valuable alternative to classical sampling and Ames test in specific settings. I therefore support the publication of this manuscript.
ljne 11, Abstract: "Ames test" is maybe more common term than "Salmonella assay" for the abstract.
It was changed
Supplement: VBC: please explain abbreviation, suggest no capital letters for glucose, nitroflurene sodium azide..
The VBC (Vogel-Bonner medium) was included into the text as requested. The capital letters were avoided, too.